# Alkali cation-π interactions in aqueous systems, modulating supramolecular stereoisomerism of nanoscopic metal-organic capsules

Paul Wix[1], Swetanshu Tandon [1], Sebastien Vaesen [1], Kadri Karimu[1], Jennifer S. Mathieson[2], Kane Esien[3], Solveig Felton [3], Graeme W. Watson [1] & Wolfgang Schmitt [1] ✉

Contrary to common chemical intuition, cation-π interactions can persist in polar, aqueous reaction solutions, rather than in dry non-coordinative solvent systems. This account highlights how alkali ion-π interactions impart distinctive structure-influencing supramolecular forces that can be exploited in the preparation of nanoscopic metal-organic capsules. The incorporation of alkali ions from polar solutions into molecular pockets promotes the assembly of otherwise inaccessible capsular entities whose structures are distinctive to those of common polyoxovanadate clusters in which {V=O} moieties usually point radially to the outside, shielding the molecular entities. The applied concept is exemplified by homologous $\{V_{20}\}$ and $\{V_{30}\}$ cages, composed of inverted, hemispherical $\{V_5O_9\}$ units. The number and geometrical organization of these $\{V_5O_9\}$ sub-units in these cages are associated with prevailing cation-$\pi$ interactions and competing steric effects. The stereoisomers of these resulting nano-sized objects are comparable to Alfred Werner-type structural isomers of simple mononuclear complexes in-line with fundamental coordination chemistry principles.

Supramolecular concepts impact on biology, medicine, catalysis, chemical synthesis and other areas of science[1]. The significance of this ever expanding field stems from the interplay between the weakness of non-covalent interactions (NCIs) and yet their simultaneous ubiquity to drive directed, thermodynamically controlled dynamic processes that are of importance in nature and science[2]. Cation-π interactions often remain unconsidered supramolecular forces in self-assembly processes[1–3], although being recognized to be amongst the most significant directional NCIs with bond energies varying between 5 to 80 kJ/mol, depending on the dielectric environment and species involved[4]. After the initial discovery and conceptualization by Kebarle

and Dougherty et al.[5,6], cation-π interactions have been found to play a role in dynamic biological systems[7–12], including histone-DNA gene regulators[13], as well as in inorganic and organic synthesis[14–17], catalysis[18,19], and material science[20,21]. Cation-π interactions involving non-cyclic molecules were initially thought to require stringent non-aqueous conditions and were predominantly found in organometallic systems synthesized in dry non-coordinative solvents under anaerobic conditions[22]. Early synthetic aromatic cation receptors that provided key-insights into the nature of the forces in polar solvents, were provided by preorganized cyclophanes, lariat ethers or related macrocycles[23–27]. It is particularly important to note that the binding

[1]School of Chemistry & SFI AMBER Research Centre, Trinity College Dublin, The University of Dublin, College Green, Dublin D02 PN40, Ireland. [2]School of Chemistry, University of Glasgow, Joseph Black Building, University Ave, Glasgow G12 8QQ, UK. [3]Centre for Quantum Materials and Technologies, School of Mathematics and Physics, Queen's University Belfast, BT7 1NN Belfast, UK. ✉e-mail: schmittw@tcd.ie

energy associated with cation-π interactions can indeed compensate the hydration enthalpies of cations[28], allowing the transfer of specific inorganic ions from aqueous to more hydrophobic environments - the understanding and application of these processes may be exploited for the formation of entirely new, otherwise inaccessible, supramolecular assemblies.

Coordination cages and metal-organic polyhedra represent supramolecular architectures whose well-defined 3D cavities can facilitate specific host-guest interactions, cavity-directed chemical transformations or the stabilization of highly reactive species, promoting applications for instance as sensors or catalysts[29-32]. The field has evolved whereby the rational design of advanced molecular hosts with tailored cavity sizes/geometries, specific binding sites and compositional diversity remain synthetic challenges[30,31]. Polyoxometalates (POMs) are anionic oxo-clusters of the early transition metal ions in their higher oxidation-states; the intrinsic chemical attributes of POMs, including polyoxovanadates, which find applications in areas such as catalysis, electronics, magnetism, medicine or environmental sciences[33-42], render these species as intriguing potential components or building units of molecular capsules. However, condensation reactions in aqueous vanadate systems containing the pyramidal $\{VO_5\}$ species, often result in the assembly of relatively small, closed-shell inorganic cages with inaccessible cavities. Their short, terminal, $\{d\pi–p\pi\}$-stabilized V=O bonds point radially to the outside, hampering the accessibility of metal-organic species and often requiring tailored functionalization approaches[38].

We have been interested in the functionalization of polyoxometalates (POMs) using organoarsonate and -phosphonate ligands[43-45]. Primarily, our synthetic efforts were directed towards hybrid-vanadate cages involving the linkage of pentanuclear $\{V_5O_9\}$ or related tetranuclear $\{V_4O_8\}$ units[43-45]– an area that was initially pioneered by Müller, Pope and Zubieta[46-53] and which received considerable attention through the work of Su, Wang, Zaworotko and others[54-58]. The bowl-shaped $\{V_5O_9\}$ units are well-established mixed-valent $\{V^{IV}_4V^VO_9\}$ structural units in polyoxometalate chemistry that form upon partial reduction of $V^V$ salts in $H_2O$/DMF solutions, whereby nucleophilic species like halides, $N_3^-$ or $H_2O$ can act as templates; in these units, a central $V^V$ ion with pyramidal coordination environment is surrounded by four $V^{IV}$ ions[43,44,51]. Hybrid POMs facilitate a versatile functionalization of molecular oxides through the systematic modification of the organic moieties[38], Other synthetic approaches allowed

the isolation of distinct metal-organic species[59], including interconvertible and nested Platonic and Archimedean metal-organic polyhedra[60,61].

Here we demonstrate how cation-π interactions involving alkali metal ions and aromatic moieties of di- and tri-functional phosphonates, persist in the assembly process of nanoscopic molecular $\{V_{20}\}$ and $\{V_{30}\}$ cages. The account highlights how cation-π interactions can be exploited in polyoxovanadate systems to produce fundamentally new nanoscopic metal-organic capsules. The stereochemistry of these complex supramolecular assemblies is governed by the electronic and steric influences associated with cation-π interactions. Interestingly the topologically reduced symmetries of the resulting $\{V_{30}\}$ isomers relate to those of simple Alfred Werner-type stereoisomers of mononuclear octahedral complexes, providing a link to fundamental coordination chemistry principles. The applied supramolecular approach results in capsular entities that are structurally remarkably different from classical polyoxovanadate cages in which the terminal $\{V=O\}$ moieties shield the spherical molecular entities[43-53]. Contrary to common chemical intuition, the observed cation-π interactions are observed in polar, aqueous reaction solutions and in the solid-state structures.

## Results

The title compounds crystallize upon partial reduction of sodium vanadate(V) using hydrazine hydrate as a reducing agent in $H_2O$/DMF/MeCN mixtures in the presence of the corresponding phosphonic acids and alkali metal chlorides (Fig. 1). The use of benzene-1,3-diphosphonic acid ($H_4$BDP) promotes the formation of toroidal $[M_8(Na, K)(H_2O,Cl)_2 \subset (V^VV^{IV}_4O_9)_4(BDP)_8]^{11/13-}$ complexes **M-$\{V_{20}\}$**, whilst the use of benzene-1,3,5-triphosphonic acid ($H_6$BTP) results in octahedral $[(V^VV^{IV}_4O_9)_6(BTP)_8]^{30-}$ capsules **M-$\{V_{30}\}$**, that form different stereoisomers and accommodate variable numbers of alkali metal ions, M.

### The $\{V_{20}\}$ system

The **M-$\{V_{20}\}$** structures are shown in Fig. 2. **K-$\{V_{20}\}$**, as highlighted in Fig. 2A–D, is also representative for **Rb-$\{V_{20}\}$** and **Cs-$\{V_{20}\}$**. The toroidal vanadate cages form around central cubic assemblies of alkali metal ions. In all these inner units, a central $Na^+$ or $K^+$ ion is bridged to 8 surrounding $M^+$ ions that engage in cation-π interactions with the phenyl moieties of the organic phosphonates; the alkali ions are bridged

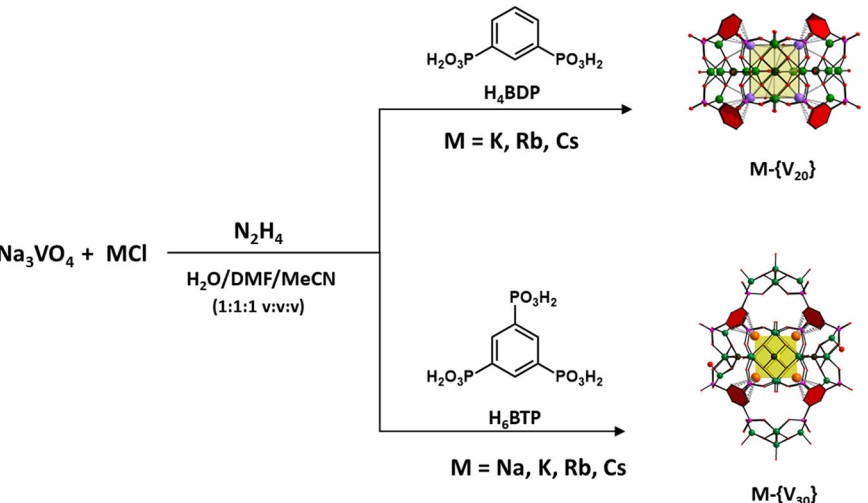

**Fig. 1 | Formation of hybrid vanadate capsules.** Partial reduction of $Na_3VO_4$ using hydrazine hydrate as a reducing agent in the presence of di- and tri-functional organophosphonate $H_4$BDP or $H_6$BTP pro-ligands and alkali metal ions M, results in the formation of **M-$\{V_{20}\}$** and **M-$\{V_{30}\}$** cages. The method was adapted from synthetic procedures that give rise to compounds with $\{V_5O_9\}$ building units. In addition, the syntheses employ $N_3^-$ ions which are thought to act as templates in the formation capsular entities which are composed of mixed-valent $\{V_5O_9\}$ units[43,44,51].

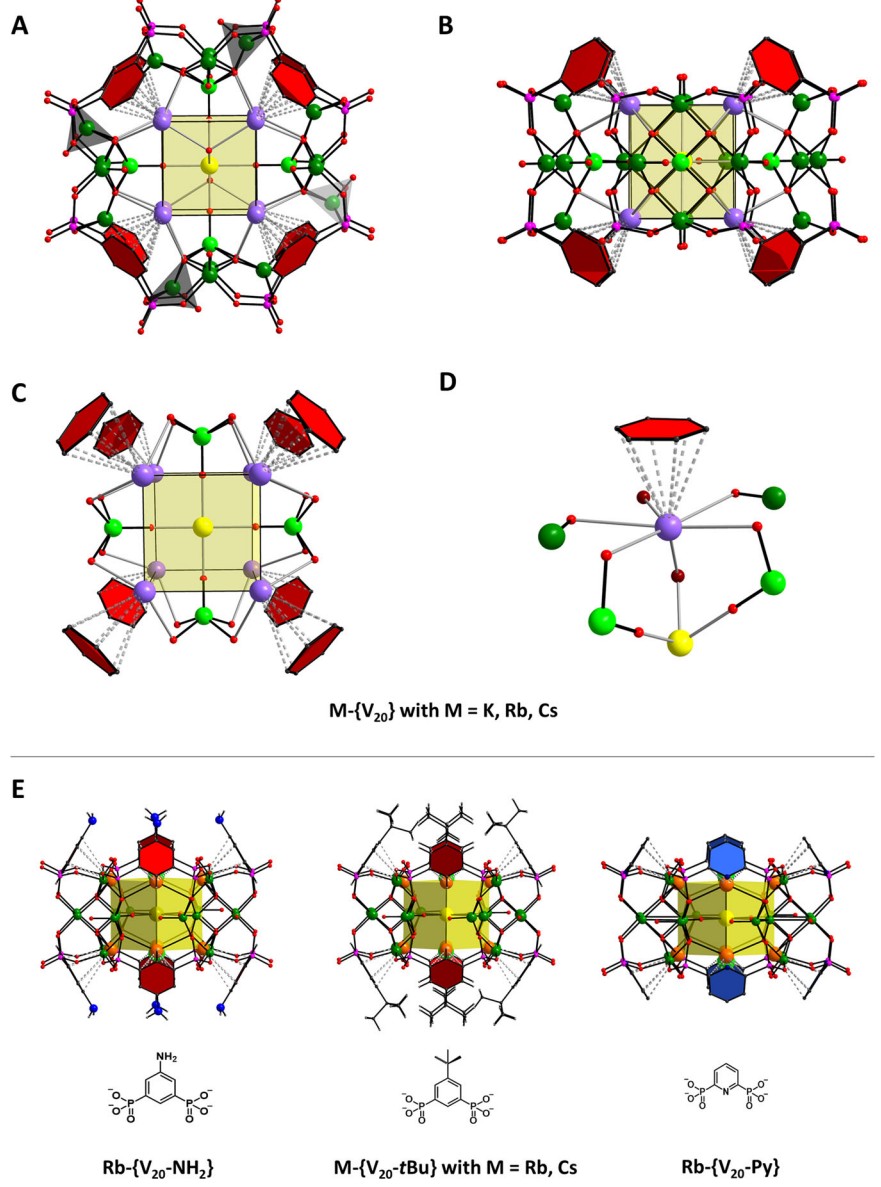

**M-{V$_{20}$} with M = K, Rb, Cs**

Rb-{V$_{20}$-NH$_2$}          M-{V$_{20}$-*t*Bu} with M = Rb, Cs          Rb-{V$_{20}$-Py}

**Fig. 2 | Structure of the toroidal {V$_{20}$} species and structural versatility.**
**A**, **B** Different perspectives of **K-{V$_{20}$}**, representative for all [M$_8$(Na,K)(H$_2$O,Cl)$_2$⊂(V$_5$O$_9$)$_4$(BDP)$_8$]$^{11-/13-}$ complexes with M = K, Rb, Cs. **C**, **D** Cation-π interactions in **K-{V$_{20}$}** and binding environment of the alkali metal ions. **E** Ligand modifications: Left: 3-Amino-1,3-benzene diphosphonate; Middle: 3-*t*Butyl-1,3- benzene diphosphonate; Right: Pyrdine-1,5-diphosphonate. Colour codes and notation: V green (for **A-D**: V$^V$ light green; V$^{IV}$ darker green); Rb orange; K purple; Na yellow; P pink; O red; N blue; C grey/black; cation-π interactions between phenyl moieties and Rb$^+$ ions are shown as intersected bonds; H-atoms are removed for clarity.

through four {V=O} oxo-groups, two H$_2$O molecules or Cl$^-$ ions that locate on the faces of a cubic inner assembly. The encapsulating hybrid-vanadate torus is composed of four {V$_5$O$_9$} units that are linked by the eight benzene-1,3-diphosphonate ligands. The V atoms in the {V$_5$O$_9$} units comprise of terminal oxo-groups and adopt typical square pyramidal coordination environments. In-line with previous reports[43,44,51], the central V ion is thought to adopt the oxidation state +V and is connected via $\mu_3$-O$^{2-}$ ligands to four outer V$^{IV}$ centers that are stabilized by *anti, anti O,O'*-bridging phosphonate moieties of the organic ligands. The infrared (IR) spectra of the compounds (Supplementary Fig. 1) give rise to signals at *ca.* 1110–1125 cm$^{-1}$ and *ca.* 1040 cm$^{-1}$ which can be attributed to the {P-O} vibrations of the coordinating phosphonate ligands while the broad signals at *ca.* 960–970 cm$^{-1}$ stem from vibrations of the {V$^V$=O} and {V$^{IV}$=O} moieties[51,62,63]. The broadness of the latter signals and observed shoulders are consistent with the mixed-valent nature of the compounds[63–65].

It is striking that in **M-{V$_{20}$}**, all 8 phenyl moieties of the BDP$^{4-}$ ligands cap the M$^+$ alkali ions and locate at the vertices of the inner cubic assemblies, building out strong cation-π interactions (Fig. 2C&D) that are characterized by distances between the M$^+$ ions and the mean plane of the phenyl rings of 3.232(4), 3.271(3) and 3.277(4) Å for **K-{V$_{20}$}**, **Rb-{V$_{20}$}**, and **Cs-{V$_{20}$}**, respectively. The remaining coordination sites of the alkali ions are provided by *oxo*-groups of vanadyl units and donor atoms of solvent molecules or anions. Interestingly, we were not able to isolate the **Na-{V$_{20}$}** homologue in which Na$^+$ ions engage in cation-π interactions, although the cation-π bond energies decrease down the alkali metal group, Na$^+$ > K$^+$ > Rb$^+$ ≈ Cs$^+$, as calculated for gas-phase systems[66]. However, in condensed systems the interactions compete with the solvation enthalpies which are expected to influence the cage formation pathways in solution. The hydration energies decrease down the alkali group whereby the smaller Na$^+$ ion reveals the highest hydration energy within the investigated

homologous series (409 kJ/mol (Na$^+$); 322 kJ/mol (K$^+$), 293 kJ/mol (Rb$^+$) and 264 kJ/mol (Cs$^+$))[67] competing with cation-π stabilization effects. This is consistent with the observation that cation-π interactions are predominantly observed for the larger alkali metal ions in nature[66], or in organometallic compounds that form in non-coordinating solvents[6,22], and in preorganized host systems that limit solvation interactions[23–27].

Organic ligand modifications and introduction of *tert*-butyl (R = *t*Bu) or amine (R = NH$_2$) moieties in the 5-position of the phenyl ring or the use of the pyridine-2,6-diphosphonic acid ligand (H$_4$BDP-Py), also produce the corresponding toroidal {V$_{20}$} polyoxovanadate structures (Fig. 2E). These syntheses underline the structural versatility of the reaction system and indicate that the homologous series of modified {M$_8$Na(Cl)$_2$⊂(V$_5$O$_9$)$_4$(BDP-R)$_8$} species, **M-{V$_{20}$-*t*Bu}**, **M-{V$_{20}$-NH$_2$}** and **M-{V$_{20}$-Py}**, are accessible. The electronic effects somewhat influence the strengths of the cation-π interactions as indicated by the observed interatomic distances (Table 1). A comparison of interatomic alkali metal distances and crystal packing diagrams of the presented compounds are provided in the Supplementary Figs. 2-8.

### Inverted vs. classical polyoxovanadates

The pentanuclear {V$_5$O$_9$} units in **M-{V$_{20}$}** as shown in Fig. 3 differ from classical polyoxovanadates or hybrid vanadates in which the hemispherical {V$_5$O$_9$}$_c$ units orientate their concave side inwards to form *closed (c)* capsular entities (Fig. 3A&C)[43,44,46–53]. Here, the formation of the inner cubic alkali core promotes a rather unusual arrangement in which the concave, *open (o)* surfaces of the concavo-convex {V$_5$O$_9$}$_o$ units point to the outside and in which the organic ligand moieties point into opposite directions. The associated steric implications of this arrangement force one {V$^{IV}$=O} vanadyl group within each of the four {V$_5$O$_9$}$_o$ units to adopt an inverted conformation[68], pointing with the *oxo*-group towards the focal point of the concave side (Fig. 3B). The associated steric restrains further force the vanadyl inversion within the four units within the {V$_{20}$} structures to be cooperative, thus leading to a clockwise or counter-clockwise arrangement of {V$^{IV}$=O} groups around the {V$_{20}$} torus (Fig. 3D). Density functional theory (DFT) calculations were carried out on the pentanuclear sub-units to investigate the energetics associated with the flipping of a {V$^{IV}$=O} vanadyl pyramid and the formation of the inverted {V$_5$O$_9$}$_o$ units. Energy minimizations were carried out on both types of the pentanuclear vanadate unit, the classical {V$_5$O$_9$}$_c$ version that gives rise to *closo* {V$_{10}$} capsules[29,30] (Fig. 3A, E) and the here-observed {V$_5$O$_9$}$_o$ arrangement in which one {V$^{IV}$=O} group is inverted pointing towards its inner focal point (Fig. 3B, F, Supplementary Fig. 9) - a position that in classical vanadates is occupied by a nucleophilic species (e.g. N$_3^-$, H$_2$O or halide ions) and that are known to template the formation of such vanadate subunits[43–45,51]. For computational easement the BDP$^{4-}$ ligands were replaced with methylphosphonate ligands. The calculations were carried out using the Gaussian09 package[69] implementing the hybrid DFT functional PBE0 in conjunction with 6-31 G(p,d,2 d) and SDDALL basis sets, to model H-, C-, N-, O- and P-atoms, and providing effective core potentials for the V atoms, respectively[70–73]. The choice of functionals and basis sets was informed by our previous experience and literature evidence demonstrating their suitability to accurately compute structural and electronic properties of related first-row transition metal complexes including vanadate species[74–76].

The models accurately reproduced the bond-distances as well as the bond-angles of both pentanuclear polyoxovanadate units (Supplementary Information). Importantly the Gibbs free energy calculations of the charge-neutral entities underline the dynamic, interconvertible nature of the two {V$_5$O$_9$} units whereby the conventional {V$_5$O$_9$}$_c$ arrangement is stabilized by *ca.* 27 kJ/mol in respect to the inverted {V$_5$O$_9$}$_o$ arrangement in the {V$_{20}$} structures. A templating nucleophile like H$_2$O adds additionally *ca.* 36 kJ/mol to the stabilization of the {V$_5$O$_9$}$_c$ unit. Contrarily, cation-π stabilization and

### Table 1 | Alkali cation-π interactions in the vanadate cages

| Cage-type M-{V$_{20}$} or M-{V$_{30}$} | Ligand | Distances (Å) between centroid of the phenyl ring and M$^+$ ion |
|---|---|---|
| K-{V$_{20}$} | H$_4$BDP | 3.232(4) |
| Rb-{V$_{20}$} | H$_4$BDP | 3.271(3) |
| Cs-{V$_{20}$} | H$_4$BDP | 3.277(4) |
| Rb-{V$_{20}$-NH$_2$} | H$_4$BDP-NH$_2$ | 3.225(3) |
| Rb-{V$_{20}$-Py} | H$_4$BDP-Py | 3.240(5) |
| Rb-{V$_{20}$-*t*Bu} | H$_4$BDP-*t*Bu | 3.164(4), 3.226(7), 3.201(5) |
| Cs-{V$_{20}$-*t*Bu} | H$_4$BDP-*t*Bu | 3.201(5), 3.237(5), 3.241(7), 3.254(7) |
| K-{V$_{30}$} | H$_6$BTP | 3.336(10), 3.403(11), 3.32(2), 3.357(12) |
| Rb-{V$_{30}$} | H$_6$BTP | 3.239(10), 3.237(10), 3.223(10), 3.262(10) |
| Cs-{V$_{30}$} | H$_6$BTP | 3.3447(2), 3.3070(1), 3.3557(2), 3.4165(1) |

Interatomic distances (Å) between the centroid of the phenyl ring of the organic phosphonate ligands and M$^+$ ions (M = K, Rb, Cs).

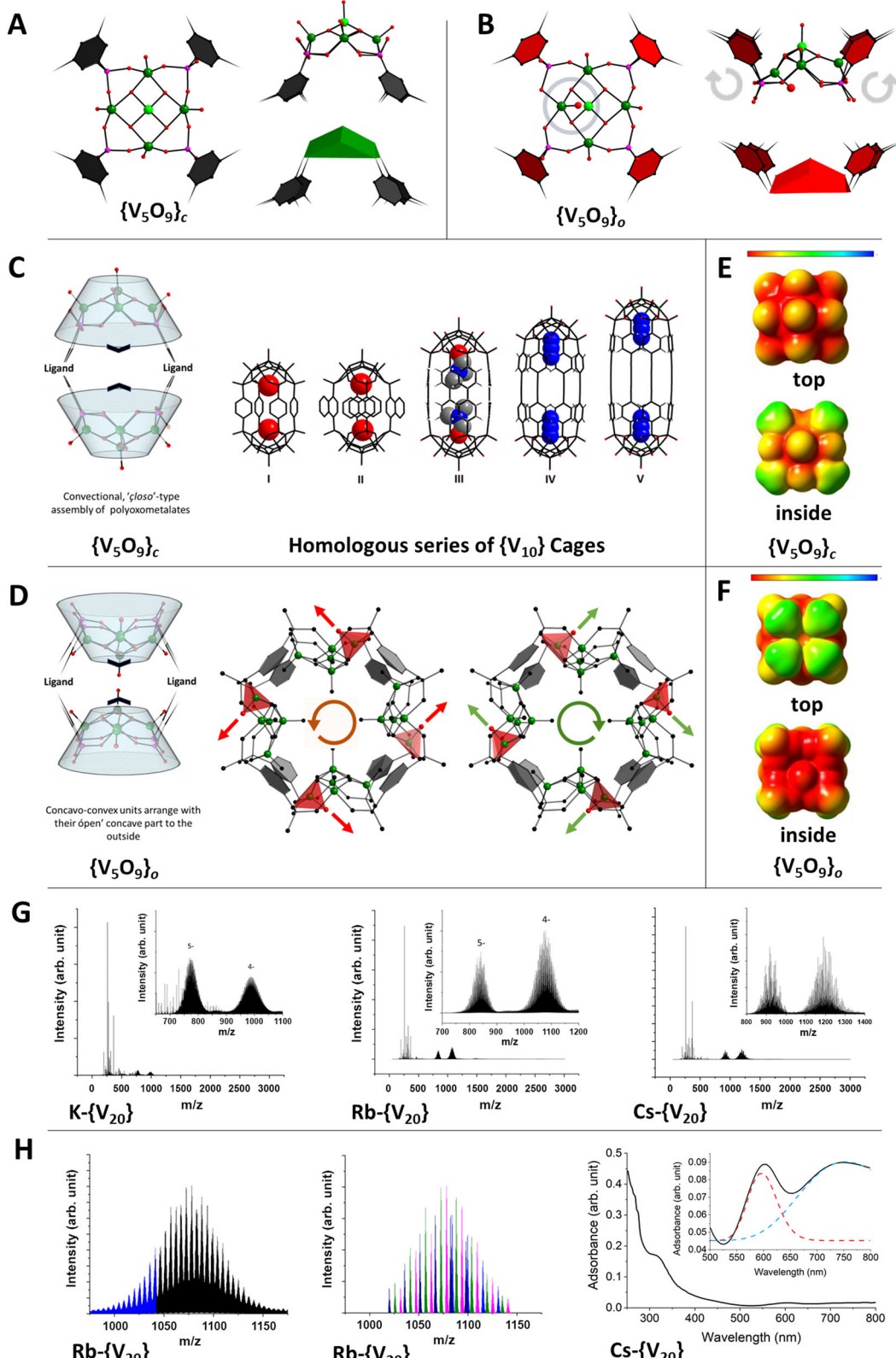

**Fig. 3 | Arrangement of {V₅O₉} units and solution-stability of the M-{V₂₀} species. A** Classical calix-type $\{V_5O_9\}_c$ unit present in hybrid and conventional polyoxovanadates; **B** Inverted $\{V_5O_9\}_o$ unit with flipped ligand arrangement and in which a $\{V^{IV}=O\}$ group points towards its focal point; **C** Conventional *closo*-type assembly in polyoxometalates and exemplar homologous series of molecular capsules I-V that form with linear organo-bisphosphonates[43,44]. **D** Unusual *open* assembly in **M-{V₂₀}** and clock and counter-clockwise arrangement of the $\{V^{IV}=O\}$ moieties. **E, F** Molecular Electrostatic Potential (MEP) map for $\{V_5O_9\}_c$ and $\{V_5O_9\}_o$, respectively. Scale: −0.51 to −0.2 e−/(r_{Bohr})[3]; **G** ESI-MS spectra of **K-{V₂₀}** (*left*), **Rb-{V₂₀}** (*center*) and **Cs-{V₂₀}** (*right*). **H** *Left* Isotopic envelope of −4 charged **Rb-{V₂₀}**, whereby all black-coloured signals derive from species that contain all alkali metal ions of the inner cavity; *Center* Simulated ESI-MS signal for **Rb-{V₂₀}**; *Right* UV-VIS spectrum (*black line*) of **Cs-{V₂₀}** in H₂O; Deconvolution of *d-d* (*red*) and intervalence charge-transfer (IVCT) bands (*blue*).

alkali/O-donor interactions involving $\{\mu_3\text{-O}\}$ or terminal $\{V^V=O\}$ groups energetically contribute to the formation of the inverted, self-templating $\{V_5O_9\}_o$ unit. Our computed energy values are in good agreement with typical guest binding energies in closed spherical vanadate capsules[77].

The molecular electrostatic potential (MEP) maps which were calculated as part of the DFT calculations, visualize distinctive charge-distribution differences between both $\{V_5O_9\}$ units (Fig. 3E, F, Supplementary Fig. 10). The inversion to $\{V_5O_9\}_o$ makes the $\{\mu_3\text{-O}\}$ electron donors, which possess high negative electrostatic potentials within the pentanuclear unit, geometrically accessible to additionally coordinate to the inner alkali metal ions. Particularly notable is the significantly increased electron density of the central $\{V^V=O\}$ moiety in the self-templated $\{V_5O_9\}_o$ unit which bridges between the inner alkali cations in $\{V_{20}\}$ structures. An interesting supramolecular effect, reported by Hayashi and co-workers, in which an umbrella-type inversion of a $\{VO_5\}$ square-pyramidal unit in a $[V_{12}O_{32}]^{4-}$ dodecavanadate results in the reversible elimination of a guest molecule, is supportive of the here-observed phenomena[68]. Both, our DFT studies and corresponding MEP analysis are substantiated by theoretical studies of donor-acceptor interactions of structurally-related square $\{V_4O_8\}$ carboxylate-stabilized complexes, demonstrating how electron-rich, nucleophilic guests facilitate charge-transfer originating from the lone-pairs of guests to the $\sigma^*$ orbitals of the terminal $\{V=O\}$ bonds[78]. The characteristics of these parent $\{V_4O_8\}$ systems support the here observed polarizability of the bowl-shaped vanadate units and the synergistic nature involving a self-templating inversion of a $\{O_4V^{IV}=O\}$ pyramid which acts as electron-donor, inducing charge-transfer to opposite-located V-O bonds that concomitantly interact with alkali ions. This intramolecular charge transfer of the O-donor of the $\{O_4V^{IV}=O\}$ groups effectively reduces the charge of the central $V^V$ center and may contribute a degree of valence delocalization in the systems.

## Solution-stability of M-$\{V_{20}\}$ cages

To substantiate whether the M-$\{V_{20}\}$ compounds exist in solution and to gain indications whether the observed cation-π interactions may prevail during the formation of the compounds, electrospray ionization (ESI) mass spectrometry and UV-vis absorption experiments of M-$\{V_{20}\}$ in aqueous solutions were conducted (Fig. 3G&H). The resulting mass spectra are characterized by sets of two well-defined isotopic envelopes between m/z = 720–1075 for K-$\{V_{20}\}$, m/z = 750–1200 for Rb-$\{V_{20}\}$ and m/z = 800–1400 for Cs-$\{V_{20}\}$. These envelopes are composed of signals resulting from 4- and 5- charged $\{M_8Na(H_2O,Cl)_2\subset(V_5O_9)_4(BDP)_8\}$ species in which the V centers adopt different oxidation states and which carry various counterions, thus demonstrating that the alkali ions in the inner cavities are an integral part of the polyoxovanadate species that exist in aqueous solutions (see also Supplementary Figs. 11–32; Supplementary Tables 1-8.). The UV-VIS spectra in $H_2O$ are characterized by signals that arise from ligand-centered transitions in the UV region, LMCT $\pi(O) \rightarrow d(V)$ bands at *ca.* 320 nm and weaker $d(V) \rightarrow d(V)$ transitions at *ca.* 610 nm (Fig. 3H, right)[51]. The broadness of the signal at *ca.* 750 nm arises from inter-valence charge transfer transitions and is typical for class II mixed-valent $V^V$-$V^{IV}$ systems[65,79]. It is noteworthy that all M-$\{V_{20}\}$ compounds can be recrystallized which further substantiates their solution stability.

## The $\{V_{30}\}$ system

A synthetic approach to modify this reaction system involves the introduction of an additional phosphonate functionality at the phenyl ring to form benzene-1,3,5-triphosphonic acid ($H_6$BTP). The geometrical alignment of the third phosphonate group is predetermined by the formation of the toroidal $\{V_{20}\}$ assembly and results in the formation of $[(V_5O_9)_6(BTP)_8]^{30-}$ $\{V_{30}\}$ capsules that incorporate various alkali ions (Fig. 4, Supplementary Figs. 2, 7 & 8,). These M-$\{V_{30}\}$ cages with octahedral molecular topology formally result through the

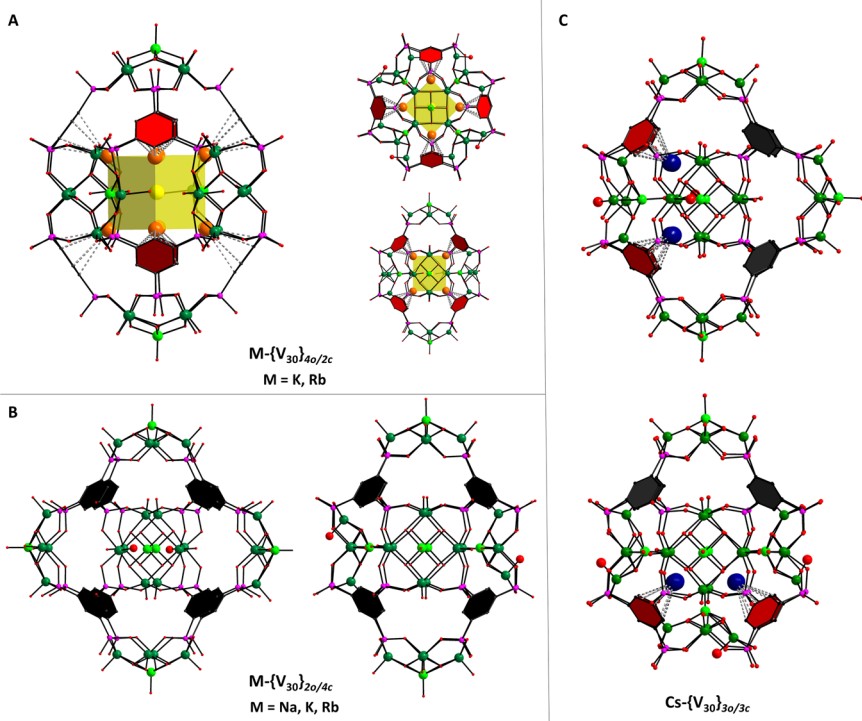

**Fig. 4 | Structural variation of $\{V_{30}\}$ capsules. A** Structure of M-$\{V_{30}\}_{4o/2c}$, M = K, Rb; **B** Structure of M-$\{V_{30}\}_{2o/4c}$ M = Na,[#] K, Rb; **C** Structure of Cs-$\{V_{30}\}_{3o/3c}$; Structures are shown from different perspectives, respectively. Colour code: $V^V$ light green; $V^{IV}$ green; Rb orange; Cs blue; Na yellow; P pink; O red; C grey; cation-π interactions between phenyl moieties and alkali ions are shown as intersected bonds; phenyl H-atoms and Cs-O bonds removed for clarity. [#]*Note*: In the Na-$\{V_{30}\}$ structure the inversion/flipping of the V=O moieties in the $\{V_5O_9\}_o$ units only occurs at very low propensity.

attachment of two additional {V₅O₉} units, above and below the {V₂₀} torus. The obtained structures consist of variable numbers of *closed* inwards, and *open* outwards-oriented {V₅O₉} units forming different geometrical isomers.

Depending on the nature of the alkali halide, isomeric **M-{V₃₀}** species can be characterized. Within the potassium/vanadate system, two stereoisomers, composed of corresponding *trans*-located {V₅O₉} units, **K-{V₃₀}**$_{2c/4o}$ and **K-{V₃₀}**$_{4c/2o}$ co-crystallize (Fig. 4A, B). In **K-{V₃₀}**$_{2c/4o}$, structural features of the toroidal {V₂₀} vanadates, including the inner cubic alkali assemblies and left or right-handed {V=O} alignments within the torus, are maintained. In addition, two inwards-oriented {V₅O₉}$_c$ units cap the {V₂₀} ring structures. The interatomic distances associated with the cation-π interactions are consistent with those observed in the **M-{V₂₀}** system (Table 1; M = K, Rb, Cs). Contrarily, in the second **K-{V₃₀}**$_{4c/2o}$ isomer four *closed* inwards-oriented {V₅O₉}$_c$ units are connected to form equatorial ring moieties, which are capped by two *open* outwards-oriented {V₅O₉}$_o$ units to form the capsular entities. The four central hemispherical {V₅O₉}$_c$ units do not provide suitable binding sites within the inner cavity of the cages and cubic alkali assemblies are not observed for **K-{V₃₀}**$_{4c/2o}$. Although only a low-resolution structure for **Rb-{V₃₀}** was obtained, the X-ray data indicates the structural resemblance to **K-{V₃₀}**, with the two *trans*-stereoisomers co-crystallizing. However, the latter systems differ distinctively from the homologous **Na-{V₃₀}** species, whose structure exclusively features the **Na-{V₃₀}**$_{4c/2o}$ isomer, which does not contain any encapsulated Na⁺ ions in the cavity that engage in cation-π interactions. The X-ray structure was refined as **Na-{V₃₀}**$_{4c/2o}$, treating the interior electron density of the {V₃₀} cage as disordered solvent molecules. Its well-resolved structural model and associated geometrical parameters provide adequate comparison to the isomers in **K-{V₃₀}** and allow the unambiguous determination of the oxidation states of the vanadium ions within the pentanuclear half-capsules by Bond Valence Sum calculations (Supplementary Note 2, Supplementary Tables 9–15). From a crystallographic point of view, the structures of **K-{V₃₀}** were refined with partial occupancy of the two isomers. Considering the nature of this positional disorder, the observed cation-π interactions may conceptionally be associated with either isomer, **K-{V₃₀}**$_{2c/4o}$ and/or **K-{V₃₀}**$_{4c/2o}$. The crystal structure of **Na-{V₃₀}** clearly addresses this ambiguity with **Na-{V₃₀}**$_{4c/2o}$ showing no signs of any cation-π interactions. While it is possible that disordered sodium ions may be present within the cavity of **Na-{V₃₀}**$_{4c/2o}$, none of them are to be found in the vicinity of phenyl rings as in the **K-{V₃₀}**, confirming the association of the cation-π interactions within {V₂₀} rings in the **K-{V₃₀}**$_{2c/4o}$ isomer. The absence of the cation-π interactions in **Na-{V₃₀}** and the predominant formation of **Na-{V₃₀}**$_{4c/2o}$ is consistent with the absence of a toroidal cation-π stabilized **Na-{V₂₀}**$_{4o}$ species and increasingly competing solvation energies of the smaller Na⁺ ions within the homologous alkali series. Hence, the structure provides valuable insights into the self-assembly mechanism that governs the formation of the different cage species in the presence of alkali ions.

**Cs-{V₃₀}** is obtained when 1 eq. of Na₃VO₄ is partially reduced in a H₂O/MeCN/DMF mixture in the presence of <1 eq. benzene-1,3,5-triphosphonic acid and 3 eq. of CsCl. Although the resulting crystals were small and weakly diffracting, the X-ray data allowed us to locate the large Cs⁺ ions in the cage and establish the connectivity of a **Cs-{V₃₀}**$_{3o/3c}$ cage structure as shown in Fig. 4C. The entity consists of three *closed* inwards, and three *open* outwards-oriented {V₅O₉} units forming a nanoscopic cage whose structure formally corresponds to that of a simple octahedral *meridional* stereoisomer. Interestingly the cavity accommodates only four Cs⁺ ions which are engaged in cation-π interactions with BTP⁶⁻ ligands that stabilise the outwards arranged, non-classical {V₅O₉}$_o$ subunits. Thus, this half of the structure is consistent with the structural features of the **M-{V₂₀}** species. The remaining part of the supramolecular cage reveals the

classical POM-type structure, whereby all terminal {V=O} bonds of the hemispherical {V₅O₉}$_c$ units point radial to the outside. This part of the structure is not stabilised by alkali metal ions. The structural model of **Cs-{V₃₀}**$_{3c/3o}$ is consistent with the other **M-{V₂₀}** and **M-{V₃₀}** species, underlining the synergy between two neighbouring *open* outwards-oriented {V₅O₉}$_o$ arrangements and the cation-π interactions.

For all structures and consistent with the DFT calculations, the self-templating {Vᵛ=O} moieties that point towards the focal points of the {V₅O₉}$_o$ units increase the electron density of the central {Vᵛ=O} and μ₃-O-donor groups which become coordinatively accessible and can bridge between alkali ions located in the cavity of the cages. Preliminary ESI mass spectrometry analyses also indicate that the {V₃₀} cages are stable in aqueous solutions (Supplementary Fig. 32).

## Discussion

The here applied synthetic methodology takes advantage of distinctive binding modes of the organophosphate ligands that stabilize the pentanuclear vanadate unit and their ability to form hydrophobic pockets that promote the binding of alkali metal ions in competition with their hydration or solvation in polar reaction mixtures. The interatomic distances between the alkali ions in the {V₂₀} and {V₃₀} derivatives and isomers are consistent, demonstrating how these cages provide a favorable binding environment for these main group ions (Supplementary Fig. 2). The observed assembly process provides a synthetic avenue to unconventional inverted polyoxovanadates in which the {V=O} moieties point to the inside rather than shielding capsular *closo* cages to the outside. This inversion is expected to be associated with altered chemical attributes of the structurally defined nanoscopic objects, facilitating potential substrates unrestricted access the inner convex surfaces of the {V₅O₉}$_o$ moieties and the binding sites of its constituent square pyramidal {O=VO₄} polyhedra. This feature, including the charge-transfer through the flipped {V=O} moieties, shows resemblance with the undulated {O=VO₄}ₙ layers in bulk V₂O₅, which is the industrially most important vanadium compound with applications as acid-base and redox catalysts[80]. Hence, the observed structural attributes render the isolated species as potentially interesting supramolecular hosts for guest interactions and potential applications in homogeneous catalysis.

As exemplified for the **M-{V₂₀}** species, the steric hindrance that is associated with the prevailing cation-π interactions and the formation of the unusual open arrangement of the concavo-convex units, is mitigated by the cooperative, directional inversion of one vanadyl {Vᴵⱽ=O} group in adjacent {V₅O₉}$_o$ units (Fig. 3D).

The isolated {V₃₀} structures exemplify the geometrical duality between a cube and an octahedron. Possible combinations of assembling six {V₅O₉}$_c$ and {V₅O₉}$_o$ units to form octahedral {V₃₀} cages can be conceptualized by a cube-net representation as shown in Fig. 5A. In a {V₃₀} system consisting of six, 3-dimensionally inter-connected {V₅O₉} units, the directional mechanism of steric avoidance, as observed for **M-{V₂₀}** species, is not applicable to all units. The triangular faces of an octahedral structure prohibit a uniform directional flipping of the {Vᴵⱽ=O} groups of neighboring {V₅O₉}$_o$ units. Hence, one observes the formation of distinct stereoisomers involving the two possible forms of the {V₅O₉} units.

Figure 5B provides a pictographic summary of the isolated toroidal **M-{V₂₀}** structures and stereoisomeric **M-{V₃₀}** cages that incorporate various different alkali ions. The six possible di- and tri-substituted stereoisomers of **M-{V₃₀}** can be exemplified using simple structural reductions that compare to Alfred Werner-type isomers of mononuclear octahedral complexes. The two possible *trans* isomers are represented by {V₃₀}$_{2c/4o}$ and {V₃₀}$_{4c/2o}$ within the homologous series of the alkali ions. The smaller Na⁺ ions show a reduced ability to engage in cation-π interactions and preferentially form the latter {V₃₀}$_{4c/2o}$ structure over {V₃₀}$_{2c/4o}$ whose formation is governed by interactions between the phenyl rings heavier alkali ions. The structure

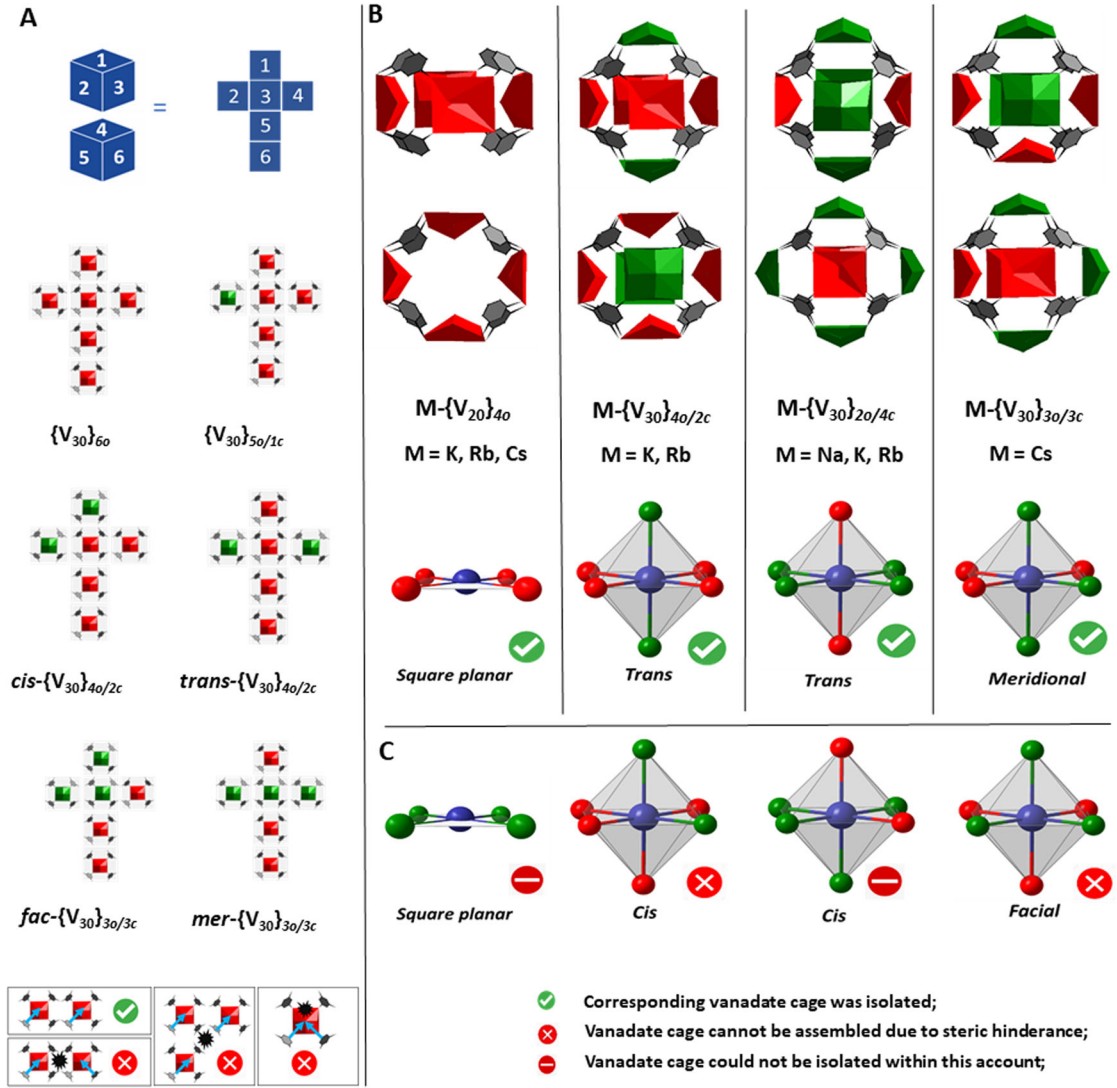

**Fig. 5 | Comparison of the cage topologies with the stereoisomers of simple mononuclear complexes. A** Cube-net representation of possible combinations of placing two types of $\{V_5O_9\}$ units around a central cube. Further four isomers are inverse structures that result from the switching $\{V_5O_9\}_o$ and $\{V_5O_9\}_c$ units in the representations. Steric constraints that limit the formation of some isomers are shown below. **B** Structural representations of the isolated vanadate cages and topological structural reduction to square planar complexes and octahedral isomers. **C** Isomers that are not attainable due to steric hindrance or were not isolated within this account; Color Code: *green* classical calix-type$\{V_5O_9\}_c$ unit; *red* inverted $\{V_5O_9\}_o$ unit.

of **Cs-$\{V_{30}\}_{3c/3o}$** is indicative of the formation of a *meridional* supramolecular isomer whilst the **M-$\{V_{20}\}$** structures with M = K, Rb and Cs represent the square planar arrangement. Consistently, within the investigated solvent system, the Na$^+$ ions do not appear to form the **Na-$\{V_{20}\}$** structure, in-line with the increased solvation energies that may compete with the cation-$\pi$ interactions that stabilize the $\{V_{20}\}$ ring structures.

It should be considered that the $\{V_5O_9\}_o$ arrangements in the $\{V_{20}\}$ complexes, enables the central $\{V=O\}$ moiety to bridge between M$^+$ ions which engage in cation-$\pi$ interactions. This may not be consistently applicable to the possible $\{V_{30}\}$ structures. Figure 5C further highlights the di- and tri-substituted octahedral *cis* and *facial* stereoisomers that may be considered, however cannot form due to steric restrains associated with the concerted inversion of the $\{V^{IV}=O\}$ groups of neighboring $\{V_5O_9\}_o$ units (see also Fig. 5A, bottom). The possible *cis*-**M-$\{V_{30}\}_{4c/2o}$** and $\{V_{30}\}_{6c}$ assemblies and the more conventional *closed* $\{V_{20}\}_{4c}$ structure (Fig. 5C), were not observed within reaction system. The latter assembly with square-planar molecular topology is realized as a part of the $\{V_{30}\}_{2o/4c}$ and carboxylate-stabilized structures[81].

In summary, we show how cation-$\pi$ interactions can govern the assembly of nanoscopic metal-organic vanadate capsules. We demonstrate that these non-covalent interactions can prevail in aqueous polyoxometalate systems giving rise to the formation of homologous nanoscopic molecular entities in which hemispherical vanadate units are inverted and whose structures differ significantly from conventional, classical POM cages. The reported **M-$\{V_{20}\}$** species are solution-stable as investigated by mass spectrometry and can further be structurally modified through ligand substitution and introduction of functional groups. The larger **M-$\{V_{30}\}$** species accommodate variable numbers of alkali ions and form different isomers as a result of steric constraints and competing cation-$\pi$ interactions. The supramolecular isomerism of the resulting nano-sized objects is influenced by these cation-$\pi$ interactions and is comparable to simple Alfred Werner-type structural isomers of simple mononuclear complexes providing simplified analogies to fundamental coordination chemistry principles. Future studies will further investigate the synthesis and resolution of various **M-$\{V_{30}\}$** stereoisomers in solid-state and solution. Our structural and computational analyses underline the synergistic nature of the inversion

of $\{O_4V^{IV}=O\}$ pyramids which act as electron-donors and the charge-transfer that is associated with the alkali ion binding and cation-$\pi$ interactions. The nature of these assembly processes and associated energies may give rise to dynamic interconversions between the observed structures and their 3D structural variation involving different $\{V_5O_9\}_c/\{V_5O_9\}_o$ arrangements may potentially translate to selective supramolecular recognition phenomena that underpin applications in catalysis or sensing. The applied synthetic approach may be applicable to other structurally related vanadate building units with 4- and 5-membered ring motifs and other poly-oxometalates that form spherical capsular structures with polarizable terminal metal-oxo bonds. The related organoarsonate ligand system in which $\{As-O\}$ bond lengths closely match those of $\{V^V-O\}$ bonds, is expected to produce varied hybrid molecular entities that are distinctive from the here reported species[45,52].

## Methods

### Materials
All chemicals were purchased from commercial sources and were used without further purification. Water was deionized and distilled before use. The organophosphonate ligands were prepared following literature methods[82], or as described in the Supplementary Note 1.

### Syntheses of M-$\{V_{20}\}$, M = K, Rb, Cs
For the preparation of all toroidal M-$\{V_{20}\}$ structures, 0.070 g $Na_3VO_4$ (0.381 mmol, 1 eq.), 0.070 g $NaN_3$ (1.08 mmol, 2.83 eq.) were dissolved in the presence of 3 eq. of the corresponding alkali metal chloride, MCl, in 3 mL $H_2O$. The resulting reactant mixture was then added to a solution of 0.050 g benzene-1,3-diphosphonic acid (H$_4$BDP, 0.210 mmol, 0.55 eq.) in 1 mL DMF, prior to the addition of 1 mL MeCN. The stirred yellow solution was heated to 70 °C before the pH value was adjusted to pH = 7 using concentrated aqueous HCl. After the addition of 16 μL of $N_2H_4{\cdot}H_2O$, the solution turned green and the pH value was re-adjusted to pH = 7 using the former acid. After stirring for 1 min at 70 °C, the solution was allowed to cool down to room temperature. The green solution was divided into 1 mL portions. Approx. 0.5 mL MeCN was allowed to diffuse into these samples. Green crystals suitable for single-crystal X-ray diffraction experiments, formed after *ca.* 2 days. For further synthetic details and physicochemical characterizations see Supplementary Notes 1 & 2, Supplementary Figs. 1, 11–21, 33 & 34 and Supplementary Tables 1–4.

### Syntheses of Rb-$\{V_{20}$-Py$\}$, M-$\{V_{20}$-tBu$\}$ and M-$\{V_{20}$-NH$_2\}$, M = K, Rb
For the synthesis of the $\{V_{20}\}$ derivatives, H$_4$BDP, was substituted by 5-*tert*-butyl-1,3-benzenediphosphonic acid (H$_4$BDP-*t*Bu), 3,5-anilinedi-phosphonic (H$_4$BDP-NH$_2$) acid or pyridine-2,6-diphosphonic acid (H$_4$BDP-Py), respectively For further synthetic details and physicochemical characterizations see Supplementary Notes 1 & 2, Supplementary Figs. 1, 22-31 and Supplementary Tables 5-8.

### Syntheses of M-$\{V_{30}\}$, M = Na, K, Rb, Cs
The syntheses of M-$\{V_{30}\}$ structures employ analogous preparative methods using benzene-1,3,5-triphosphonic acid (H$_6$BTP, 0.066 g, 0.208 mmol) as organic ligand. Here, the addition of $N_2H_4{\cdot}H_2O$ and stirring for 1 hour at 70 °C, results in the formation of dark green precipitates (see Supplenetary Note 1). The latter solids were isolated at room temperature by filtration, dissolved in 3 mL $H_2O$ and residual insoluble particles were again removed by filtration. Following the addition of 0.5 mL DMF to the filtrate, green crystals that formed reproducibly were separated manually within 1-3 weeks. The nature of the multi-step method to crystallize the M-$\{V_{30}\}$ structures generally resulted in relative low yields of single-crystals which were suitable X-ray diffraction experiments. For further details and physicochemical characterizations see Supplementary Notes 1 & 2, Supplementary Figs. 1, 32 and 33. Crystallographic details for all compounds are provided in Supplementary Tables 16–20.

## Data availability
Crystallographic data for the structures reported in this article have been deposited at the Cambridge Crystallographic Data Centre under deposition numbers CCDC 2350523-2350532 and 2382883. Copies of the data can be obtained free of charge via www.ccdc.cam.ac.uk/structures/. Data supporting the findings, including synthetic procedures and analyses are contained within the publication and its Supplementary Information file. Additional source data are available from the corresponding author on request.

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

## Acknowledgements

This research was funded by Science Foundation Ireland (13/IA/1896 (W.S., P.W., S.T., S.V.); 12/RC/2278_P2 (W.S.)) and the European Research Council (ERC SUPRAMOL CoG 2014–647719 (W.S., S.V.) and the European Innovation Council (EIC Transition AIRINMOTION 101136924 (W.S., S.V.). P.W. and S.T. acknowledge fellowships from the Irish Research Council (IRC, GOIPG/2015/2713 (P.W.), GOIPG/2015/2952 (S.T.)). S.F. and K.E. acknowledge support from DfE (Department for the Economy, Northern Ireland) through grant USI 108. The authors thank Prof. Lee Cronin, University of Glasgow for facilitating the mass spectrometry measurements and Dr. Brendan Twamley for the help with the crystallographic analyses and refinements.

## Author contributions

W.S. conceived the project, was in involved in data interpretation and wrote the manuscript. P.W. conducted the syntheses, characterizations and data analyses including X-ray crystallography. S.T. and G.W.W. conducted the DFT calculation. K.K. was involved in the synthesis. S.V. performed data analyses for the characterization of the compounds. J.S.M. conducted mass spectrometry analyses. K.E. and S.F. performed SQUID measurements. All authors contributed to the writing of the manuscript.

## Competing interests

The authors declare no competing interests.
