## [Peer Review file · Nature Communications]

Alkali Cation- π Interactions in Aqueous Systems, Modulating Supramolecular Stereoisomerism of Nanoscopic Metal-Organic Capsules

Corresponding Author: Professor Wolfgang Schmitt

Version 0:

Reviewer comments:

Reviewer #1

(Remarks to the Author)

Dear editor,

Sorry for the late reply. I read the manuscript carefully with great interest. Though the manuscript is mainly structural, I think it is interesting and can be accepted for publication in your journal after some revisions.

1. For the checkcif files of 504439_0_data_set_8980135_sc1518, 504439_0_data_set_8980131_scj518 and 504439_0_data_set_8980137_scg518, the reported formula and the calculated formula of these compounds are different, please give an explanation.
2. The authors used BVS to calculate the valence of vanadiums, please give the formula they used in the supplementary material.
3. Actually, the IR can be used to determine the valence of vanadiums, please see DALTON TRANSACTIONS, Volume 42, Issue15, Page 5247-5251.
4. Please give the bond lengths between alkali metals in the vanadium cage in the manuscript.

Reviewer #2

(Remarks to the Author)

In the manuscript entitled "Alkali Cation- π Interactions in Aqueous Systems, Modulating Supramolecular Stereoisomerism of Nanoscopic Metal-Organic Capsules" Schmitt et al. report a series of V20 and V30 capsules containing symmetric arrays of alkali metal ions which are held in place by cation- π interactions. The authors first describe the structures of the V20 clusters accompanied by theoretical calculations and ESI-mass spectrometry and then describe the V30 structures and discuss the influence of steric hindrance and the interplay of intramolecular interactions on the formation of different stereoisomers.

The authors stress the importance of the cation- π interactions, that they persist in aqueous solution and that they lead to the observation of inverted structures. This is the major message of this manuscript which would be of significance to a general science audience and to workers in this particular field. However, the question is whether the authors have presented convincing arguments to adequately substantiate this.

There are several deficiencies in both the data and the discussion of the results which make it difficult to recommend this work in its current form for publication in Nature Communications.

The concerns are presented below.

1. Crystallography:

5 of the 10 reported structures have B and/or A alerts in their checkcifs which have not been addressed. In particular, the last two structures are of very low resolution and with unacceptably high R1 and wR2 (e.g. R1 22% and wR2 53%) factors. This Cs4V30 structure has only been refined with isotropic thermal parameters for all atoms including the metal ions. In the Rb8V30 structure the wide range of thermal parameters for the V atoms suggests that the disorder of these has not been adequately modelled. These structures would benefit from being remeasured using a high brilliance source such as a synchrotron. Given the size and complexity of the structures any crystallographer would sympathise with the challenges

posed in refining these. However, although these structures are indeed interesting the poor quality of those with A alerts precludes publication.

2. Results and Discussion:

It is a puzzle how the authors can assign the apparently well-defined oxidation states of the V centres so confidently. Experience shows that it is quite possible to have non-integer oxidation states in mixed-valence compounds. Indeed, the presented UV-Vis data suggest IVCT transitions. The authors should explicitly state with appropriate references how the transitions in the absorption spectra have been assigned (page 10).

The authors state that in the DFT calculations “the BDP ligands were replaced with methylphosphonate ligands for computational easement...”. This implies they have simplified the structures by replacing each aromatic ring of the phosphonate ligands with a pair of Me groups. This requires clarification. It is not clear that this substitution is chemically justified. It is also unclear how structures with no aromatic rings can be used to quantify the cation- π interactions. The relevance of the Werner-type isomers is an attractive aspect of this manuscript. Unfortunately, two out of three structures have already been identified as being crystallographically and therefore structurally deficient. If these structures could be remeasured as suggested above, then this would lead to a well-defined and important selling point. The mass spectrometry is convincing with since the peaks and the fragments have been identified and assigned extremely well.

3. Further issues:

Some of the figures are of very poor quality and need to be reworked (e.g. Figure 4 bottom right)

In the light of these criticisms the conclusion section may need rewriting once the authors have addressed the issues outlined above.

As it stands the introduction needs a substantial rewrite since it is both repetitive and provides little insight into the importance and relevance of V-based capsules.

4. Summary and recommendation:

In summary, there is a lot potentially interesting material presented here. However, the quality, and therefore the interpretation, of some of the data need to be improved to make this contribution suitable for publication in Nature Communications.

Reviewer #3

(Remarks to the Author)

Reviewer #4

(Remarks to the Author)

This is a very nice manuscript that describes the synthesis and characterisation of a series of structurally fascinating vanadate cages stabilised by di- and tri-substituted benzene phosphonate ligands. The work highlights a new, supramolecular approach that is governed by alkali- π interactions, resulting in inverted polyoxometalates (POMs) that are distinctively different to classical POM architectures in which the V=O bonds point radial to the outside. Perhaps most interestingly is that the observed cation- π interactions persist in aqueous solution and that, despite the structural complexity, the stereoisomers of the V₃₀ cages can be related to simple octahedral cis-trans, meridional-facial isomers. This provides some degree of structural design that may be adaptable in other reaction systems. The results are highly interesting to the general chemistry community, and potentially highly impactful. The manuscript is well written/presented and I therefore recommend publication in Nature Communication subject to the minor corrections/comments/suggestions listed below which the authors can use, abuse, ignore as they see fit.

In future experiments, the authors aim to investigate the structurally related molybdates or tungstates. This seems feasible, but would it not be equally interesting to investigate variable ligand systems? Within many POM families organophosphonates are often exchangeable with organoarsonates (Mueller, Zubietta, Clearfield). Have the authors explored if similar cages can be obtained using organoarsonates?

What is the rationale of conducting the experiments in a 3-component solvent mixture rather than in a pure solvent system? The introduction should perhaps refer to other previously published ligand-stabilised vanadate structures that may be composed of other relevant vanadate oxo-units, e.g. Nature Communications 11(1), 4103, DOI:10.1038/s41467-020-17989-6; Nature Communications; 9(1), 4941, DOI: 10.1038/s41467-018-07427-z; Angew Chem Int. Ed. 2001, 40(21), 4018.

Figure 1 caption could/should be expanded to provide more detailed information on, and explanation of, the figure shown. Typo in the figure – DBP should be BDP (top right). Define what A⁺ ions are in the caption.

Figures 2, 4. Check the colour codes are correct. I think there are a few mistakes here. Purple is what? Rb = orange...also Cs/K in E?

Figure 4C right: 30/3C?

It is not particularly clear from the text/figures which ions are V(V) and which are V(IV) – the light green versus dark green colouring is rather tricky to distinguish in places. Perhaps this can be amplified a little? The authors should also provide a clearer description of this in the text and the formulae with oxidation states should be provided in the main text at least once. Could the concerted directional ‘flipping’ of the V=O bonds of adjacent {V₅O₉} units potentially result in chiral, non-superimposable geometrical isomers? Do the authors observe resolved geometrical enantiomers/isomers in the reaction

systems?

What is the role of NaN₃ in the synthesis of M-V20? Please add a comment in the SI.

There appears to be a large error in the %C in the elemental analysis of K-V30; 8.03% vs 6.73% real or typo? That said I understand that %CHN for such large species is often unreliable/not particularly accurate.

Can any assignments of the peaks in the IR spectra be made? If so, please add.

Insets in the top right-hand corners of the PXRD data are very small and hard to read. Same thing for the numbers on the axes.

Large variations in R1. Some more refinement details would be welcome in the XRD section of the SI.

Provide a reference for the BVS calculations. Values of 3.32 and 3.34 assigned as 4+ in V20tB-Rb (Table 6) could use an explanation (perhaps in the caption). Same for 4.55/4.56/4.51 assigned as V5+ for Rb-V20 and Cs-V20 (Table 5-6). That said, I believe the assignments are correct.

The SI Supplementary Notes suggest that SQUID measurements were performed, but no data are given. Same with UV-Vis and TGA. Either provide these data in the SI or remove references to them in the text. The mixed-valent nature of the of the V5O9 units involving S=1/2 V(IV) centres could give rise to interesting magnetic properties and SQUID measurements of selected compounds could provide interesting, preliminary insights into this, but are not a pre-requisite for accepting the manuscript.

Typographical errors

Abstract, line 8: "in which the" repeated.

Page 2, line 29: "...the formation of..."

Page 7, line 13: The spelling of 'polyxometalates' should be corrected.

Page 20, line 19: "envelopes".

Figure 4: "silver".

Page 13, line 22: four or two Cs?

Page 14, line 5: delete "and".

Page 14, line 28: "can be XXX by a cube.."

Page 18, line 24: "reproducibly". Delete "and".

Page 15, line 1: "dimensionally"

Page 18, line 2: "were" rather than "was".

Version 1:

Reviewer comments:

Reviewer #1

(Remarks to the Author)

[Note from the Editor: Reviewer #1 made comments to the Editor only and thinks that the revision is sufficient and that the manuscript can be accepted in its current form.]

Reviewer #2

(Remarks to the Author)

The authors are thanked for responding to the points which they have done in completely satisfactory manner. This manuscript can now be accepted for publication in Nature Communications.

Reviewer #3

(Remarks to the Author)

Reviewer #4

(Remarks to the Author)

The authors have done a very good job in the revised manuscript and have addressed all of the questions posed by the referees. I am happy to accept the manuscript in its current form.

Re: Response to Reviewers' Comments

We thank the reviewers for their very valuable input and the time they invested. We very much appreciated the comments/suggestions and modified the manuscript as requested (please see below).

Reviewer #1:

Reviewer: I read the manuscript carefully with great interest. Though the manuscript is mainly structural, I think it is interesting and can be accepted for publication in your journal after some revisions. **Response:** We thank the reviewer for taking the time and very much appreciate the interest in our manuscript.

Reviewer: For the checkcif files of 504439_0_data_set_8980135_sc1518, 504439_0_data_set_8980131_scj518 and 504439_0_data_set_8980137_scg518, the reported formula and the calculated formula of these compounds are different, please give an explanation. **Response:** We thank the reviewer for highlighting this issue. We corrected the differences and we now ensure that the reported and calculated formulae are identical. We note that all the compounds contain a large number of constitutional water molecules and counterions which could not be located. This gives rise to a discrepancies between the crystallographically determined structures and compound formulae.

Reviewer: The authors used BVS to calculate the valence of vanadiums, please give the formula they used in the supplementary material. **Response:** We fully agree with the reviewer and included the formula for the BVS analyses. In addition, we provide the relevant references and parameters in the Supplementary Information file. Please see page 10 in the SI document.

Reviewer: Actually, the IR can be used to determine the valence of vanadiums, please see DALTON TRANSACTIONS, Volume 42, Issue15, Page 5247-5251. **Response:** Again, we thank the reviewer for this suggestion. The suggested publication is relevant, very interesting and supportive of the mixed-valent nature of the $\{V_5O_9\}$ units in the cages. We added short paragraphs in the manuscript and in the SI, providing an assignment of the relevant IR bands. We refer to the suggested publication and other supporting publications which also include the $\{V_5O_9\}$ unit. We now state in the manuscript: *The infrared spectra of the compounds (Supplementary Information) give rise to signals at ca. 1110-1125 cm^{-1} and ca. 1040 cm^{-1} which can be attributed to the {P-O} vibrations of the coordinating phosphonate ligands while the broad signals at ca. 960-970 cm^{-1} stem from vibrations of the $\{V^V=O\}$ and $\{V^IV=O\}$ moieties.^{51, 62, 63} The broadness of the latter signals and observed shoulders are consistent with the mixed-valent nature of the compounds.⁶³⁻⁶⁵* Please also note the additional text in the SI (see SI page 12).

Reviewer: Please give the bond lengths between alkali metals in the vanadium cage in the manuscript. **Response:** This is a very good suggestion. We now provide an overview over the bond lengths between alkali metals in the vanadium cages in the SI (see Supplementary Figure 2) and refer to this overview in the manuscript. We now state in the manuscript (page 16, line 2): *The interatomic distances between the alkali ions in the $\{V_{20}\}$ and $\{V_{30}\}$ derivatives and isomers are consistent demonstrating how these cages provide a favourable binding environment for these main group ions (Supplementary Fig. 2).*

Reviewer #2:

Reviewer: In the manuscript entitled "Alkali Cation- π Interactions in Aqueous Systems, Modulating Supramolecular Stereoisomerism of Nanoscopic Metal-Organic Capsules" Schmitt et al. report a series of V20 and V30 capsules containing symmetric arrays of alkali metal ions which are held in place by cation- π interactions. The authors first describe the structures of the V20 clusters accompanied by theoretical calculations and ESI-mass spectrometry and then describe the V30 structures and discuss the influence of steric hindrance and the interplay of intramolecular interactions on the formation of different stereoisomers.

The authors stress the importance of the cation- π interactions, that they persist in aqueous solution and that they lead to the observation of inverted structures. This is the major message of this manuscript which would be of significance to a general science audience and to workers in this particular field. However, the question is whether the authors have presented convincing arguments to adequately substantiate this.

There are several deficiencies in both the data and the discussion of the results which make it difficult to recommend this work in its current form for publication in Nature Communications. The concerns are presented below. **Response:** We thank the reviewer for the comments and judgement. We took the concerns on-board to improve the manuscript (please see below).

Reviewer: 1. Crystallography: 5 of the 10 reported structures have B and/or A alerts in their checkcif which have not been addressed. In particular, the last two structures are of very low resolution and with unacceptably high R1 and wR2 (e.g. R1 22% and wR2 53%) factors. This Cs4V30 structure has only been refined with isotropic thermal parameters for all atoms including the metal ions. In the Rb8V30 structure the wide range of thermal parameters for the V atoms suggests that the disorder of these has not been adequately modelled. These structures would benefit from being remeasured using a high brilliance source such as a synchrotron. Given the size and complexity of the structures any crystallographer would sympathise with the challenges posed in refining these. However, although these structures are indeed interesting the poor quality of those with A alerts precludes publication. **Response:** We agree with the referee and improved the quality values of some structures, addressed some alerts or now provide explanations these. Importantly, we were able to include a new high-quality Na-{V₃₀} structure which provides further insight into the assembly process and substantiates the conclusion of the manuscript.

{V₃₀} cages and new Na-{V₃₀} structure: Following the reviewer's comments, we thought about the synthetic procedure and were able to obtain good-quality crystals of Na-{V₃₀} which resulted in a well-refined, additional {V₃₀} structure that substantiates our work. The additional X-ray structure represents the *trans* isomer Na-{V₃₀}_{4c/2o}. Its well-resolved structural model and associated geometrical parameters provide a direct comparison to the two isomers in K-{V₃₀}. Both structures, Na-{V₃₀} and K-{V₃₀}, are clearly resolved and characterised by R1 values of 0.0683 and 0.0944, respectively. Na-{V₃₀} confirms the isomer assignment in K-{V₃₀}. Most interestingly, the absence of the cation- π interactions in Na-{V₃₀} and the predominant formation of Na-{V₃₀}_{4c/2o} is consistent with the absence of a toroidal cation- π stabilized Na-{V₂₀}_{4o} species and increasingly competing solvation energies of the smaller Na⁺ ions within the homologous alkali series. The new structure provides insights into the self-assembly mechanism. While it is possible that disordered sodium ions may be present within the cavity of Na-{V₃₀}_{4c/2o}, none of these can be found in the vicinity of phenyl rings as in the K-{V₃₀} structures, confirming the association of the cation- π interactions within {V₂₀} rings in the K-{V₃₀}_{2c/4o} isomer. The new Na-{V₃₀} structure clearly supports the claims of the manuscript whereby minimal disorder allows for unambiguous determination of the oxidation states in the pentanuclear sub-units via BVS analyses.

We agree that the R1 values of the two Rb and Cs-{V₃₀} cages were high. The {V₃₀} structures have low diffraction intensities due to the large quantities of disordered constitutional solvent molecules and counterions located in the voids. The disorder associated with two superimposed isomers and the 'disorder within the disorder' further increases the complexity and results in diffuse electron density which is very difficult to model.

We further worked on these systems. We obtained an improved, anisotropically-refined model for Cs-{V₃₀}. The R1 value converged at 0.1268 (previously >0.22). Although the diffraction data is of low intensity at higher 2 Θ angles and the structure is of lower resolution, the structure undoubtedly allowed us to locate the heavy Cs⁺ ions within the cage, providing a realistic structural model which we can stand over. Unfortunately we were not able to improve the Rb-{V₃₀} structure due to twinning of these crystals. As the structure contains the same *trans* isomers as K-{V₃₀}, we could simply remove this compound from the manuscript without any changes to the conclusions. However, we feel that for chemical consistency, it is important to include the compound (*Note:* It is actually very rare in the literature that series of compounds can be prepared across the entire alkali series due to their significantly different atomic radii). Whilst we now provide a more detailed description of the Na-{V₃₀} and K-{V₃₀} structures in the revised manuscript, we removed the emphasis from the Rb- and Cs-{V₃₀} structures. Associated claims were kept to a minimum and we removed the image highlighting the detailed binding geometry in Cs-{V₃₀}. We clearly state the limitation of the latter two structures, but providing models that are consistent with the other structures. We also removed any statement in relation to the number of structures we report. We added additional crystallographic details in the SI document (please see SI page 8).

{V₂₀} cages: The R1 values of the 7 {V₂₀} structures now only vary between 5.07% and 8.14%. The structures in combination with the mass spectrometry data clearly elucidate the role of the prevailing cation- π interactions in the solid and solution phase. The B alerts for **Rb-{V₂₀-tBu}** and **Cs-{V₂₀-tBu}** stem from the C-C bond precision associated with the disordered *tert*-butyl groups. An additional B alert for **Rb-{V₂₀-tBu}** probes for missing H-atoms in H-bonds, but often occurs for POM structures. It results from the short terminal {V=O} moieties (due to $d\pi$ - $p\pi$ contributions) and close contacts in the typical square pyramidal V coordination environments. The observed disorder further contributes to this B alert. For **Cs-{V₂₀-tBu}**, we reduced Θ_{\max} slightly during the data integration to account for the negligible diffraction intensity at higher Θ angles (influenced by the large number of constitutional solvent molecules and counterions). We provide reasonings for these alerts in the cif-files.

We included the following text in the manuscript: *Although only a low-resolution structure for Rb-{V₃₀} was obtained, the X-ray data indicates the structural resemblance to K-{V₃₀}, with the two trans-stereoisomer pairs co-crystallizing.*

However, the latter systems differ distinctively from the homologous Na-{V₃₀} species, whose structure exclusively features the Na-{V₃₀}_{4c/2o} isomer, which does not contain any encapsulated Na⁺ ions in the cavity that engage in cation- π interactions. The X-ray structure was refined as Na-{V₃₀}_{4c/2o} treating the interior electron density of the {V₃₀} cage as disordered solvent molecules. Its well-resolved structural model and associated geometrical parameters, provide adequate comparison to the isomers in K-{V₃₀} and allow the unambiguous determination of the oxidation states of the vanadium ions within the pentanuclear half-capsules by Bond Valence Sum calculations (see BVS calculations in the Supplementary Information). From a crystallographic point of view, the structures of K-{V₃₀} were refined with partial occupancy of the two isomers. Considering the nature of this positional disorder, the observed cation- π interactions may conceptually be associated with either isomer, K-{V₃₀}_{2c/4o} and/or K-{V₃₀}_{4c/2o}. The crystal structure of Na-{V₃₀} clearly addresses this ambiguity with Na-{V₃₀}_{4c/2o} showing no signs of any cation- π interactions. While it is possible that disordered sodium ions may be present within the cavity of Na-{V₃₀}_{4c/2o}, none of them are to be found in the vicinity of phenyl rings as in the K-{V₃₀}, confirming the association of the cation- π interactions within {V₂₀} rings in the K-{V₃₀}_{2c/4o} isomer. The absence of the cation- π interactions in Na-{V₃₀} and the predominant formation of Na-{V₃₀}_{4c/2o} is consistent with the absence of a toroidal cation- π stabilized Na-{V₂₀}_{4o} species and increasingly competing solvation energies of the smaller Na⁺ ions within the homologous alkali series. Hence, the structure provides valuable insight into the self-assembly mechanism that governs the formation of the different cage species in the presence of alkali ions.

Reviewer: 2. Results and Discussion: It is a puzzle how the authors can assign the apparently well-defined oxidation states of the V centres so confidently. Experience shows that it is quite possible to have non-integer oxidation states in mixed-valence compounds. Indeed, the presented UV-Vis data suggest IVCT transitions. The authors should explicitly state with appropriate references how the transitions in the absorption spectra have been assigned (page 10 in the manuscript). **Response:** We agree with this comment. The transitions in the adsorption spectra are consistent with previously reported vanadate clusters. We added the following paragraph providing the relevant references (page 11 line 21): *The UV-VIS spectra in H₂O are characterized by signals that arise from ligand-centered transitions in the UV region, LMCT $\pi(O) \rightarrow d(V)$ bands at ca. 320 nm and weaker $d(V) \rightarrow d(V)$ transitions at ca. 610 nm (Fig. 3H, right).⁵¹ The broadness of the signal at ca. 750 nm arises from inter-valence charge transfer transitions and is typical for class II mixed-valent V^V-V^{IV} systems.^{65,79}*

In addition, the IR spectra are consistent with the observed mixed-valent nature of the compounds, as are the included representative magnetic data for the {V₂₀} system. The {V₅O₉} unit with a central V^V ion is a well-known mixed-valent unit that occurs in numerous vanadate species (see references and work of A. Müller). The significant structural disorder within the {V₅O₉} units and the disorder due to superimposed {V₃₀} isomers, influence the V-O bond precisions and the associated Bond-Valence-Sum analysis. We now clearly state this in the SI (see SI page 8; crystallographic section). In the absence of the disorder (see Na-V₃₀), the oxidation sates are generally more clearly defined. Further, we agree with the judgement of the referee acknowledging the intramolecular charge-transfer within the systems, which effectively influences the charge distribution and the oxidation states. Indeed, the DFT calculations (see also the Molecular Electrostatic Potential maps) clearly demonstrate how electron-density is transferred to the central V^V=O moiety. We now state in the manuscript: *This intramolecular charge transfer of the O-donor of the {O₄V^V=O} groups effectively reduces charge of the central V^V centre and may contribute a degree of valence delocalization in the systems.*

Reviewer: The authors state that in the DFT calculations “the BDP ligands were replaced with methylphosphonate ligands for computational easement...”. This implies they have simplified the structures by replacing each aromatic ring of the phosphonate ligands with a pair of Me groups. This requires clarification. It is not clear that this substitution is chemically justified. It is also unclear how structures with no aromatic rings can be used to quantify the cation- π interactions. **Response:** We thank the reviewer for this comment and agree that this aspect was not clearly described in the manuscript. The DFT calculations were carried out on the pentanuclear sub-units to investigate the energetics associated with the ‘flipping’ of a {V^{IV}=O} vanadyl pyramid and the formation of the inverted the {V₅O₉}_o units in comparison to the conventional {V₅O₉}_c unit. We now clearly state this at the start of the relevant section (page 8 line 25). Due to the large number of atoms, it was not realistically feasible to calculate the entire cluster including the cation- π interactions.

Reviewer: The relevance of the Werner-type isomers is an attractive aspect of this manuscript. Unfortunately, two out of three structures have already been identified as being crystallographically and therefore structurally deficient. If these structures could be remeasured as suggested above, then this would lead to a well-defined and important selling point.

Response: We thank the reviewer for the comment, which motivated us to add a new **Na-V₃₀** structure to the manuscript (see previous response). The two possible *trans* isomers **{V₃₀}_{2c/4o}** and **{V₃₀}_{4c/2o}** are confirmed by the **Na-{V₃₀}** and **K-{V₃₀}** structures. The **M-{V₂₀}** structures with M = K, Rb and Cs represent the square planar arrangement. We improved the structure of **Cs-{V₃₀}_{3c/3o}** and although of lower resolution, the location of the heavy Cs⁺ ions confirms the existence of the *meridional* supramolecular isomer. As previously described, the new **Na-{V₃₀}** structure is interesting as it sheds light onto the assembly process. The smaller Na⁺ ions show a reduced ability to engage in cation- π interactions, in-line with the increased solvation energies that compete with the cation- π interactions. Hence, Na⁺ ions form **{V₃₀}_{4c/2o}** and not the cation- π stabilised **Na-{V₂₀}** structure. We modified the Results & Discussion and Conclusion sections appropriately. We stated the limitations of the **Rb-{V₂₀-tBu}** and **Cs-{V₂₀-tBu}** structures. Their descriptions and associated claims were kept to a minimum. We could remove **K-{V₃₀}** from the manuscript without any changes to the conclusions. However, we feel that for chemical consistency, it is important to mention the existence of this compound (see above).

Reviewer: The mass spectrometry is convincing since the peaks and the fragments have been identified and assigned extremely well. **Response:** We thank the reviewer for this comment.

Reviewer: Some of the figures are of very poor quality and need to be reworked (e.g. Figure 4 bottom right). **Response:** We apologise for this. The MS-Word/Adobe file compression changed the resolution of some of the images. We addressed this issue. In addition, we can provide all image at high-resolution (≥ 300 dpi).

Reviewer: In the light of these criticisms the conclusion section may need rewriting once the authors have addressed the issues outlined above. As it stands the introduction needs a substantial rewrite since it is both repetitive and provides little insight into the importance and relevance of V-based capsules. **Response:** We altered the conclusions considering the corrections and inclusion of the new structure. We ensured that the claims are substantiated by data. We feel that it is important to highlight the prevailing cation- π interactions which govern the supramolecular assembly and synthetic concept. We deleted repetitive phrases and re-worked the introduction, and as suggested, we added a paragraph referring to the importance and relevance of V-based capsules (please see introduction).

Reviewer: Summary and recommendation: In summary, there is a lot potentially interesting material presented here. However, the quality, and therefore the interpretation, of some of the data need to be improved to make this contribution suitable for publication in Nature Communications. **Response:** We thank the reviewer for this judgment and improved the manuscript according to the very constructive comments.

Reviewers #3 and 4:

Reviewers: This is a very nice manuscript that describes the synthesis and characterisation of a series of structurally fascinating vanadate cages stabilised by di- and tri-substituted benzene phosphonate ligands. The work highlights a new, supramolecular approach that is governed by alkali- π interactions, resulting in inverted polyoxometalates (POMs) that are distinctively different to classical POM architectures in which the V=O bonds point radial to the outside. Perhaps most interestingly is that the observed cation- π interactions persist in aqueous solution and that, despite the structural complexity, the stereoisomers of the V₃₀ cages can be related to simple octahedral cis-trans, meridional-facial isomers. This provides some degree of structural design that may be adaptable in other reaction systems. The results are highly interesting to the general chemistry community, and potentially highly impactful. The manuscript is well written/presented and I therefore recommend publication in Nature Communication subject to the minor corrections/comments/suggestions listed below which the authors can use, abuse, ignore as they see fit. **Response:** We really thank the reviewers for these comments and are pleased that they rate the results so highly.

Reviewers: In future experiments, the authors aim to investigate the structurally related molybdates or tungstates. This seems feasible, but would it not be equally interesting to investigate variable ligand systems? Within many POM families organophosphonates are often exchangeable with organoarsonates (Müller, Zubieta, Clearfield). Have the authors explored if similar cages can be obtained using organoarsonates? **Response:** We appreciate this comment and share the opinion that organophosphonates and organoarsonate are structurally closely related and it will be very interesting to further investigate this particular vanadium/arsonate ligand system in the future. We added a sentence referring to this aspect at the end the manuscript where we discuss the general applicability of the synthetic concept and future work. We indeed did some previous work on a corresponding p-aminophenyl arsonate vanadate system. The work supports the structures isolated by the groups Müller, Zubieta,

Clearfield. Generally, the V^V-O bond lengths are very similar to As-O bonds in organoarsonates. So arsonates have a tendency to replace the central V^V ions in the {V₅O₉} units and are expected to lead to different structures.

Reviewers: What is the rationale of conducting the experiments in a 3-component solvent mixture rather than in a pure solvent system? **Response:** It has previously been established by Achim Müller and others that the mixed-valent {V^{IV}₄V^VO₉} units form in H₂O/DMF solutions whereby nucleophilic species like N₃⁻ can act as templates (see ref 51). The addition of acetonitrile affects the crystallisation of the compounds. We added following sentence to the manuscript: *The bowl-shaped {V₅O₉} units are well-established mixed-valent {V^{IV}₄V^VO₉} structural units in polyoxometalate chemistry that form upon partial reduction of V^V salts in H₂O/DMF solutions, whereby nucleophilic species like halides, N₃⁻ or H₂O can act as templates; in these units a central V^V ion with pyramidal coordination environment is surrounded by four V^{IV} ions.*^{43,44,51}

Reviewers: The introduction should perhaps refer to other previously published ligand-stabilised vanadate structures that may be composed of other relevant vanadate oxo-units, e.g. Nature Communications 11(1), 4103, DOI:10.1038/s41467-020-17989-6; Nature Communications; 9(1), 4941, DOI: 10.1038/s41467-018-07427-z; Angew Chem Int. Ed. 2001, 40(21), 4018. **Response:** We thank the reviewers and agree. We included these relevant publications in the manuscript and altered the introduction appropriately.

Reviewers: Figure 1 caption could/should be expanded to provide more detailed information on, and explanation of, the figure shown. Typo in the figure – DBP should be BDP (top right). Define what A+ ions are in the caption. Figures 2, 4. Check the colour codes are correct. I think there are a few mistakes here. Purple is what? Rb = orange...also Cs/K in E? Figure 4C right: 30/3C? **Response:** We thank the reviewers for these comments and apologise for omissions and errors. We corrected the figure captions for Figures 1,2 & 4 and updated the colour codes. We fully agree with the suggestion and expanded the caption of Figure 1.

Reviewers: It is not particularly clear from the text/figures which ions are V(V) and which are V(IV) – the light green versus dark green colouring is rather tricky to distinguish in places. Perhaps this can be amplified a little? The authors should also provide a clearer description of this in the text and the formulae with oxidation states should be provided in the main text at least once. **Response:** We agree with the reviewers and highlight the oxidation states using a clearer colour scheme and provide formulae to clarify the oxidation states. The {V₅O₉} unit itself is a well-established moiety in the polyoxovanadate chemistry being part of numerous oxo-clusters. In these structures a central V^V centre is surrounded by four V^{IV} centres. We appreciate the comment and clarify this in the manuscript stating: *... in these units a central V^V ion with pyramidal coordination environment is surrounded by four V^{IV} ions.*^{43,44,51}

Reviewers: Could the concerted directional ‘flipping’ of the V=O bonds of adjacent {V₅O₉} units potentially result in chiral, non-superimposable geometrical isomers? Do the authors observe resolved geometrical enantiomers/isomers in the reaction systems? **Response:** The reviewers are correct. The directional ‘flipping’ of the V=O bonds could result in chiral non-superimposable isomers. However so far, we only isolated racemic structures.

Reviewers: What is the role of NaN₃ in the synthesis of M-V₂₀? Please add a comment in the SI. **Response:** This is a very good comment. NaN₃ stems from the traditional syntheses of polyoxovanadate capsules, as reported by Achim Müller (see also reference 51; *Angew. Chem. Int. Ed.* **34**, 779-781 (1995)). It is thought that N₃⁻ acts a nucleophile and template for the formation of the bowl-shaped {V₅O₉} unit. We added a sentence in the manuscript to explain this (see above).

Reviewers: There appears to be a large error in the %C in the elemental analysis of K-V30; 8.03% vs 6.73% real or typo? That said I understand that %CHN for such large species is often unreliable/not particularly accurate. **Response:** We appreciate that the reviewers highlight this issue. We re-calculated the %CHN analysis, considering crystallisation water molecules (Now: 7.49% vs 6.73%).

Reviewers: Can any assignments of the peaks in the IR spectra be made? If so, please add. **Response:** We appreciate this comment and assigned the most prominent signals in the IR spectra. The spectra are further supportive of the mixed-valent nature of the compounds. To this end, we added a short sentence in the manuscript (see also reviewer 1) and the following text in the SI: *The signals at ca. 1110-1125 cm⁻¹ and ca. 1045 cm⁻¹ can be attributed to the {P-O} vibrations of the coordinating phosphonate ligands while the broad signals at ca. 960-970 cm⁻¹ stem from vibrations of the {V=O} and {V^{IV}=O} moieties.^[S25-S27] The broadness of the latter signal and observed shoulders are consistent with the mixed-valent nature of the compounds.^[S27-S29] The broad signals around 3370 cm⁻¹ are due to O-H vibrations of H-bonded water molecules and water molecules that bind to the counterions. The bending {H-O-} vibrations of these constitutional H₂O molecules appear at ca. 1640-1660 cm⁻¹. {V-O-V} bending vibrations are expected to contribute to bands <650cm⁻¹.^[S25-S29]*

Reviewers: Insets in the top right-hand corners of the PXRD data are very small and hard to read. Same thing for the numbers on the axes. **Response:** We fully agree with the reviewers and provide improved images.

Reviewers: Large variations in R1. Some more refinement details would be welcome in the XRD section of the SI. **Response:** We agree with this comment and now provide refinement details in the XRD section of the SI (please see page 8). The R1 values vary across the reported structures as a result of the variable diffraction intensities which are influenced by the diffuse electron density of the surrounding counterions and large numbers of disordered crystallisation water molecules whose positions could not be resolved. The R1 values of the 7 {V₂₀} structures now only vary between 5.07% and 8.14%. We were able to add a well-refined structure of Na-{V₃₀} to the manuscript. Na-{V₃₀} and K-{V₃₀} are clearly resolved and characterised by R1 values of 0.0683 and 0.0944, respectively. Further, we obtained an improved structural model for Cs-{V₃₀} whose R1 value converged at 0.1268. We stated the limitations of the Rb-{V_{20-tBu}} and Cs-{V_{20-tBu}} structures and removed emphasis from these structures.

Reviewers: Provide a reference for the BVS calculations. Values of 3.32 and 3.34 assigned as 4+ in V20tB-Rb (Table 6) could use an explanation (perhaps in the caption). Same for 4.55/4.56/4.51 assigned as V5+ for Rb-V20 and Cs-V20 (Table 5-6). That said, I believe the assignments are correct. **Response:** We agree with the reviewer and included the formula for the BVS analyses. In addition, we provide the relevant references in the SI (please see page 10). The assignment of the oxidation states is consistent with literature data for the {V_{5O₉}} unit which is a general building unit of various previously reported polyoxovanadates. The observed structural disorder within the {V_{5O₉}} units of the cages and the disorder due to superimposed {V₃₀} isomers, influence the V-O bond precisions and the associated Bond-Valence-Sum analysis. We now state this in the SI (see page 8; crystallographic details).

Reviewers: The SI Supplementary Notes suggest that SQUID measurements were performed, but no data are given. Same with UV-Vis and TGA. Either provide these data in the SI or remove references to them in the text. The mixed-valent nature of the of the V5O9 units involving S=1/2 V(IV) centres could give rise to interesting magnetic properties and SQUID measurements of selected compounds could provide interesting, preliminary insights into this, but are not a pre-requisite for accepting the manuscript. **Response:** We thank the reviewers and corrected the table of content for the SI. The UV-Vis data refers to the data presented in Figure 3H (right). The reviewers are correct that the V-O connectivity in the {V_{5O₉}} units can give rise to interesting magnetic properties. We added the susceptibility data and field-dependence of the magnetisation for a representative {V₂₀} system to the SI. The data is in good agreement with 4 S=1/2 V^{IV} centres within the {V_{5O₉}} units.

Reviewers: Typographical errors

Abstract, line 8: "in which the" repeated.

Page 2, line 29: "...the formation of..."

Page 7, line 13: The spelling of 'polyxometalates' should be corrected.

Page 20, line 19: "envelopes".

Figure 4: "silver".

Page 13, line 22: four or two Cs?

Page 14, line 5: delete "and".

Page 14, line 28: "can be XXX by a cube.."

Page 18, line 24: "reproducibly". Delete "and".

Page 15, line 1: "dimensionally"

Page 18, line 2: "were" rather than "was".

Response: We thank the reviewers. We apologise for the typos and corrected these.

Note: We further did some minor changes to the manuscript/SI to comply with the editorial/formatting guidelines of *Nature Communications* (e.g. we changed the order of the data within the Supplementary Information and cross-reference the data in the manuscript according to the journal guidelines).

Re: Response to Reviewers' Comments

We appreciate the positive response and we are pleased that the referees are in agreement with our corrections. We really thank the reviewers for their efforts, valuable inputs and the time invested.